# Speaker-Independent Spectral Enhancement for Bone-Conducted Speech

Liangliang Cheng [1], Yunfeng Dou [2], Jian Zhou [1,*], Huabin Wang [1] and Liang Tao [1]

1   School of Computer Science and Technology, Anhui University, Hefei 230601, China; e20201132@stu.ahu.edu.cn (L.C.); wanghuabin@ahu.edu.cn (H.W.); taoliang@ahu.edu.cn (L.T.)
2   Anhui Finance & Trade Vocational College, Hefei 230601, China; dyunfeng7933@163.com
*   Correspondence: jzhou@ahu.edu.cn; Tel.: +86-152-5510-3835

**Abstract:** Because of the acoustic characteristics of bone-conducted (BC) speech, BC speech can be enhanced to better communicate in a complex environment with high noise. Existing BC speech enhancement models have weak spectral recovery capability for the high-frequency part of BC speech and have poor enhancement and robustness for the speaker-independent BC speech datasets. To improve the enhancement effect of BC speech for speaker-independent speech enhancement, we use a GANs method to establish the feature mapping between BC and air-conducted (AC) speech to recover the missing components of BC speech. In addition, the method adds the training of the spectral distance constraint model and, finally, uses the enhanced model completed by the training to reconstruct the BC speech. The experimental results show that this method is superior to the comparison methods such as CycleGAN, BLSTM, GMM, and StarGAN in terms of speaker-independent BC speech enhancement and can obtain higher subjective and objective evaluation results of enhanced BC speech.

**Keywords:** bone-conducted; speech enhancement; generative adversarial networks; speech reconstruction

## 1. Introduction

In the daily communication process, the background noise of the surrounding environment has a great impact on it, especially on the noisy roadside, near construction sites, train waiting rooms, and other places. The military communication field also needs to remove all kinds of noises. Therefore, improving the voice quality is very important for improving the quality of life. Environmental noise not only affects voice communication in daily life but also needs to be removed in military communication, fire rescue communication, and other fields. In the field of military communications, many places need to use voice to transmit information, and voice quality is crucial. If the voice quality is poor, the transmitted information may not be obtained correctly, resulting in very significant losses. Speech enhancement is an important technical means of reducing noise interference in speech communication. At present, speech enhancement technology has made great progress. Traditional speech enhancement methods [1,2] and depth-learning-based enhancement methods [3,4] emerge endlessly, which can achieve good enhancement effects when dealing with stationary noise. However, when the noise environment is very complex, the effect of existing speech enhancement methods will be greatly reduced. Ordinary air conduction (AC) microphones transmit information by capturing voice signals in the air, so they are very vulnerable to various environmental noises. However, it is precisely because of the unique transmission channel of its speech signal that the speech signal captured by the bone-conducted (BC) equipment can completely shield the background noise, separate the noise from the transmitted speech information from the sound source, and prevent the noise from interfering with the speech signal.

To fundamentally solve the problem of environmental noise [5], people began to use BC microphones for speech communication in some strong noise environments, such as military aircraft and tanks, where the background noise in the cockpit is very strong. In addition, the BC communication equipment is small in size, easy to carry, and can communicate normally in water. It can also be used as speech recognition under strong noise. Generally, the BC microphone with good universality mainly collects signals from sensors. The sensors are close to the skin, which can sense the vibration signals of the bones driven by human speech. Then, the collected signals are converted into formats to form BC speech signals. Different from traditional AC microphones, BC microphones only collect vibration information and will not be affected by background noise. They shield noise interference from the sound source to obtain relatively pure speech signals. This is also an important feature of BC microphones in practical applications and has important application value in complex noise environments.

BC speech can effectively resist the interference of environmental noise; however, due to the change in the sound transmission path and the limitation of the human voice generation mechanism, the BC microphone will lose some signals, especially important high-frequency voice signal information, which makes the comfort experience of the BC microphone for users unfriendly. High-frequency information of BC speech is lost, some unvoiced syllables are missing, and the sense of hearing is dull and not clear enough, so the speech intelligibility is low, and it is difficult to directly apply to normal communication [5]. These shortcomings also seriously restrict the development of BC communication.

Compared with the acoustic characteristics of AC speech, the biggest problem of BC speech is the loss of high-frequency information components. Different from the multi-sensor fusion enhancement algorithm, blind enhancement of BC speech only has BC speech information in the enhancement phase and does not need to be supplemented by AC speech features. At the same time, this process is called Blind Restoration [6]. Therefore, the difficulty of blind enhancement is to infer and recover high-frequency information based on limited information in the middle and low-frequency bands.

At present, BC speech enhancement methods include traditional methods and depth learning methods. Traditional BC speech enhancement methods are based on the Gaussian mixture model, the least mean-square error method, and other statistical methods. The traditional blind BC speech enhancement method analyzes the spectrum characteristics of BC speech from many aspects, finds the correlation between AC and BC speech, and lays a good foundation for the follow-up work. Recently, with the rapid development of deep learning technology, it has been widely used in various fields. Compared with traditional methods, the method based on deep learning can better learn the spectral characteristics of bone and air conduction speech and obtain a better enhancement effect [7–9]. DNN is gradually applied to learning the nonlinear conversion relationship between spectral envelope features [9,10]. Because the deep neural network has a good characterization ability for high-dimensional data, Liu et al. have used the deep denoising autoencoder (DDAE) to model the transformational relation of the high-dimensional Mel-amplitude spectrum features [8]. Changyan et al. [7] regarded the BC speech spectrum as a two-dimensional image, and proposed a bidirectional long short-term memory (attention-based BLSTM, ABBLSTM) on the basis of using the structural similarity (SSIM) loss function and achieved certain results. At present, most neural network enhancement frameworks are based on short-time Fourier characteristics. Most of the research inputs the spectrum into the network and thinks that the short-term phase is not important for speech enhancement. However, Paliwal et al. found through experiments that when the phase spectrum of the enhanced speech is effectively restored, the quality of speech enhancement will be significantly improved [11].

The BC speech enhancement method based on deep learning can infer high-frequency information by analyzing the low-frequency spectrum signal information composition of BC speech, reconstructing the full band speech, and improving the quality and intelligibility of the generated speech. However, the deep neural network currently used in BC speech

enhancement methods is difficult to fully learn the characteristics of BC speech in the case of limited BC speech samples and is not robust to speaker-independent speech data sets.

Recently, the authors of [12] used generative adversarial networks (GAN) to conduct in-depth learning and training of data through game learning of generators and discriminators. In theory, it has a strong generating ability. It has been widely used in many fields, such as image processing [13,14], video generation, speech synthesis [15], and so on. In particular, it has achieved considerable success in generating realistic images and solving complex data distribution problems [12]. Q. Pan et al. [16] proposed a method that can reconstruct the high-frequency part of BC speech without using the dynamic integration algorithm (DTW) for feature alignment, and the enhanced BC speech is clear.

Aiming at the problem of BC speech enhancement for speaker-independent systems, to pay full attention to the information of BC speech, we make full use of the existing features in the case of fewer training data. The BC speech enhancement generative adversarial networks for speaker-independent (BSEGAN-SI) method is proposed in this paper. In view of the shortcomings of the original GAN, this paper selects a reasonable generator and discriminator structure to build a BC speech blind enhancement architecture and directly learns the mapping relationship between BC speech and pure AC speech. This paper discusses and studies the blind enhancement methods of BC speech for speaker-independent systems, mainly to solve the problems of dull and unclear speech caused by serious high-frequency attenuation of BC speech and consonant loss and promote the practical popularization of BC speech technology. The model adopts convolutional encoder–decoder [17] architecture. Due to the convolutional shared parameters, the model can rapidly enhance BC speech. The main contributions of this paper are summarized as follows:

- The BSEGAN-SI method proposed in this paper does not need to align features and learns a priori knowledge from a small amount of data. By establishing the mapping relationship between BC and AC, the high-frequency components of BC speech can be recovered better without the assistance of AC speech in the enhancement phase.
- We add a spectral distance (L1 regularization factor) constraint to the generator to further reduce the training error, better recover the missing components of the BC speech spectrum and improve the enhancement effect so that the enhanced BC speech signal is more similar to the clear AC speech signal.
- The proposed BSEGAN-SI method only uses BC speech information in the enhancement phase and does not need AC speech features. Experimental results show that this method can better perform speaker-independent BC speech enhancement in limited data sets.

The structure of subsequent papers is as follows: Section 2 introduces the blind enhancement technology of BC speech; Section 3 introduces the method of speaker-independent BC speech enhancement; Section 4 carries out the simulation experiment and presents the results from the analysis; Section 5 summarizes the whole work.

## 2. Bone-Conducted Speech Blind Enhancement Technology

According to the different construction ideas of enhancement methods, this section introduces three typical BC speech blind enhancement methods.

### 2.1. Equilibrium Method

Shimamura et al. [18] first proposed the idea of the equalization method and built the system by modeling and constructing an inverse filter. The content of BC and AC speech collected in different ways is the same, but their transmission routes are changed. The equalization method considers that the transmission channel can be modeled, and the BC and AC speech can be mapped through the modeling function. As shown in the

transformation function Formula (1), the transformation from BC speech to AC speech is completed by combining the inverse transformation of the transmitted signal.

$$A(n) = B(n) * T_{\mathrm{B}}^{-1}(n) * T_{\mathrm{A}}(n) = B(n) * Y(n) \tag{1}$$

where $B(n)$ represents the BC speech signal, $A(n)$ represents the AC speech signal, $T_B(n)$ is the BC speech transmission channel function, $T_A(n)$ is the function of the AC speech transmission channel, and $Y(n)$ is the transformation function from BC speech to the AC speech transmission channel.

The speech conversion function is not limited to the time-domain signal but also can extract different speech features to build different transformation models, which are usually called equalization filters, for modeling the speech signal feature transformation. For example, the linear prediction cepstrum coefficient (LPCC) features and power spectral density features of BC speech and noisy speech are extracted for equalization filter modeling [19,20]. In the equalization method, prior AC and BC speech signal characteristics are first used to train the relevant parameters of the equalization filter because, in the enhancement phase, there is only a BC speech signal available. Finally, the BC speech signal is enhanced through the trained equalization filter. For example, Ref. [21] selects the corresponding bone- and air-conducted speech short-time amplitude spectrum to obtain an equalization filter, that is

$$\hat{C}(f) = \frac{\mathrm{Short}\ (AS_{\mathrm{ac}}(f))}{\mathrm{Short}\ (AS_{\mathrm{bc}}(f))} \tag{2}$$

where short refers to the short-time spectrum, and the filter coefficients are smoothed. Let the amplitude spectrum of the AC speech signal and BC speech signal be individually expressed as $AS_{\mathrm{ac}}(f)$ and $AS_{\mathrm{bc}}(f)$.

However, it is difficult to reconstruct the high-frequency components of speech signals using the above BC speech blind enhancement method because when the high-frequency energy of BC speech is almost zero, even if the response of $\hat{C}(f)$ at high-frequency is large, it is difficult to play the role of energy enhancement; the equalization filter $\hat{C}(f)$ designed in the literature is fixed. In fact, $\hat{C}(f)$ will change with the change in BC speech. Because the construction of the filter is related to the content of the speaker, the speaker's vocal tract, and the characteristics of solid tissue, such an equalization filter cannot be designed any better, nor can it be mathematically modeled to obtain such a complex nonlinear relationship.

*2.2. Bone-Conducted Speech Spectrum Expansion*

The spectrum spreading method assumes that the BC and AC speech have the same harmonic structures or formant structures. Through these similar waveform characteristics, the high-frequency signal loss of BC speech can be reduced without the assistance of AC speech, and the high-frequency signal components of BC speech can be derived and expanded directly from the low-frequency components of BC speech, which can achieve a certain enhancement effect and improve the hearing effect of the enhanced BC speech. According to the research, the low-frequency and high-frequency harmonic structures of BC and AC speech signals are similar [22]. In the experiment, based on this assumption, BC speech signals are recovered at high frequency, and the high-frequency harmonic structures are obtained according to the low-frequency information. According to the theory of the spectrum spreading method, some researchers soon strengthened the high-frequency signal components of the BC speech attenuation by adjusting the pole of the LP coefficient under the premise of the theory [23].

The advantage of the spectrum spreading method is that it is simple, convenient, and does not require other prior knowledge when enhancing BC speech. The idea of this algorithm has been applied to some BC electronic products. However, this algorithm, in fact, has certain theoretical limitations. Because the bone structure of the human body that generates vibration is different, BC recorded in different parts has the same formant

structure as AC voice, which does not fully conform to the actual voice characteristics. For instance, the method proposed by Bouserhal et al. [22] has little enhancement effect on consonants without harmonic structure. Rahman et al. conducted relevant research, and the enhancement theory proposed is only for BC speech with a wider enhancement frequency band [23]. For example, the formant structure of the high-frequency segment of BC speech recorded by a laryngeal microphone is seriously missing. When the obtained speech frequency band is narrow, the formant expansion method cannot be used to recover high-frequency information.

### 2.3. Spectral Envelope Feature Transformation

At present, most BC speech enhancement methods use the spectral envelope conversion method. This paper also uses this method to perform blind enhancement of BC speech for speaker-independent people. Its core idea is to convert the spectral envelope characteristics of BC and AC speech [10]. The spectral envelope feature represents the source-filter model derived from speech, which considers that speech is modulated by the excitation signal (source) through the sound channel (filter). BC speech enhancement can be achieved by establishing the spectral envelope feature mapping relationship between BC and AC speech. The BC speech signals $B(n)$ and AC speech signals $A(n)$ can be defined as:

$$B(n) = E_{bc}(n) * S_{bc}(n) \tag{3}$$

$$A(n) = E_{ac}(n) * S_{ac}(n) \tag{4}$$

In Formulas (3) and (4), $bc$ and $ac$ represent BC and AC speech, respectively. The excitation of speech $S_{bc}(n)$ and $S_{ac}(n)$, respectively, represent the spectral envelope of BC and AC speech. $*$ represents the convolution operation.

The blind enhancement method of BC speech based on spectral envelope conversion includes two stages. Firstly, the spectral envelope and excitation features of AC and BC speech are extracted, respectively. Then the training model constructs the complex mapping relationship $f(x)$ between source speech and target speech features. After training the model coefficients, extract the spectral envelope and excitation features of the BC speech to be enhanced according to the data processing in the previous step, and set as $E_{bc}(t)$ and $S_{bc}(t)$, respectively, as the input to the training-completed enhancement model to estimate the feature $\hat{S}_{bc}(t)$.

$$\hat{S}_{bc}(t) = f(S_{bc}(t)) \tag{5}$$

Then synthesize the enhanced speech $\hat{B}(t)$ according to the estimated spectral envelope features and the original excitation features of the BC speech.

$$\hat{B}(t) = E_{bc}(t) * \hat{S}_{bc}(t) \tag{6}$$

## 3. A Method of Speaker-Independent Bone-Conducted Speech Enhancement

### 3.1. The Overall Framework of the Proposed BSEGAN-SI Method

The framework of the BC speech enhancement method is shown in Figure 1. In this paper, the generation countermeasure network is used as the BC speech enhancement model. The WORLD vocoder [24] is used to extract the BC and AC speech feature parameters spectrum envelope (Sp) and convert them into mel-cepstral coefficients (MCEPs). Specifically, the MCEPs are the mel-log spectral approximation (MLSA) parameters of mel-frequency cepstral coefficients (MFCCs) [25]. Then the converted *MCEPs* feature is applied to network model training to learn the mapping relationship between BC and AC speech; the fundamental frequency $F_0$ is extracted for Gaussian normalization. The main goal of the generator in this framework is to model the nonlinear mapping relationship between the source speech and target speech, learn the distribution of all semantic information data of the target speech, and recover the missing information components of the source speech.

After model training, in the blind enhancement phase, we first use the vocoder to extract the MCEPS features of BC speech to be enhanced and then input them into the trained GAN model. The $F_0$ features of BC speech are converted by logarithm gaussian normalization (LGN) [26]. According to reference [27], $Ap$ does not significantly affect speech quality, so we directly use the extracted $Ap$ features of BC speech without transformation. Finally, according to the obtained speech features, we directly use the WORLD Vocoder to quickly synthesize the enhanced BC speech.

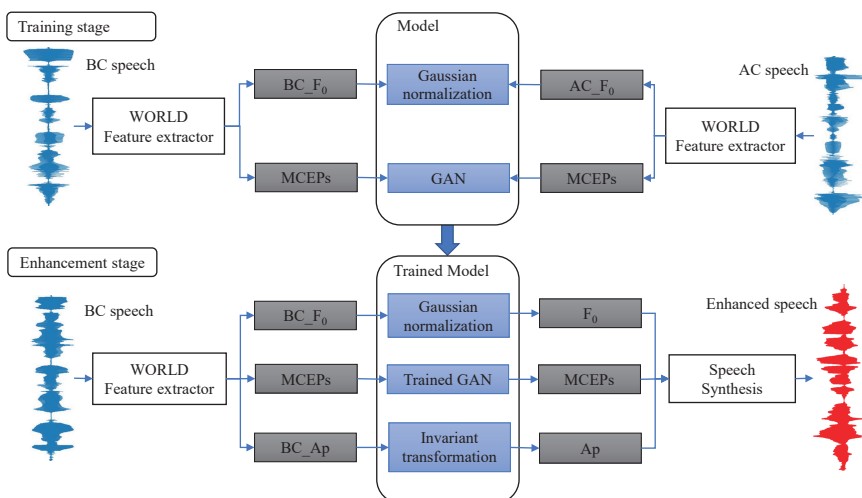

**Figure 1.** Architecture of the proposed BSEGAN-SI method.

### 3.2. The GAN Model of BC Speech Enhancement

The main problem of BC speech enhancement is the loss of high-frequency signal information and speech consonants, which requires a specific algorithm to recover the missing relevant part of speech information. However, the traditional way to infer the enhanced speech based on BC speech does not perform well in the task. With the extensive application of deep learning methods in various fields, the GAN [12] has added a binary discrimination model to improve the generation effect on the basis of a single network, which can learn high-dimensional features and complex data internal distribution, infer real samples from potential space, and can be used to generate more sample data in scenes with more missing data. Therefore, this paper uses the GAN to realize the blind enhancement task of speaker-independent BC speech. Figure 2 shows the main structure of the BSEGAN-SI method proposed in this paper. Through the power generation ability of GAN, the mapping relationship between BC and AC is established to recover part of the speech information lost by BC speech and improve the hearing quality of BC speech. According to the relevant introduction in Section 2, this paper uses the GAN to model the complex nonlinear mapping relationship between BC and AC speech signals.

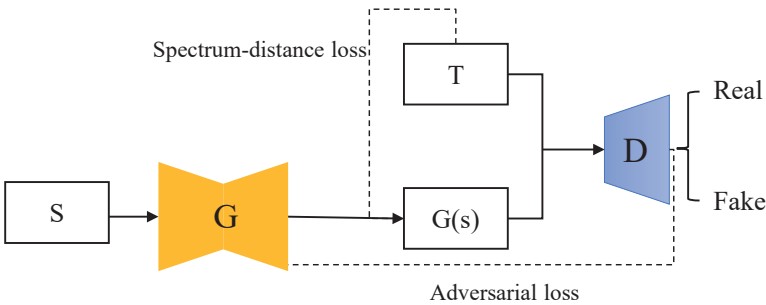

**Figure 2.** The structure of the GAN model in the BSEGAN-SI method.

We found that the sigmoid cross-entropy loss training using the original GAN is prone to instability. The LSGAN [28] method uses the loss of least squares to replace the cross-entropy loss of the original GAN and has achieved certain results in the quality of generated information data. Thus, the adversarial loss functions of discriminator and generator loss are

$$\mathcal{L}_{dis}^{adv}(D) = \frac{1}{2}\mathbb{E}_{t \sim p_{data}(T)}[(D(t) - a)^2] + \frac{1}{2}\mathbb{E}_{s \sim p_{data}(S)}[(D(G(s)) - b)^2] \tag{7}$$

$$\mathcal{L}_{gen}^{adv}(G) = \frac{1}{2}\mathbb{E}_{s \sim p_{data}(S)}[(D(G(s)) - c)^2] \tag{8}$$

where $a = 1$, $b = 0$ and $c = 1$ are fixed constants. Source BC features map $s$ and target the AC features map $t$.

In the BC speech enhancement method designed in this paper, the structure of the generator model is shown in Figure 3. The encoder–decoder [17] network framework is adopted. To enable the framework to rapidly enhance BC speech, we reasonably control the complexity of the model to achieve a faster speed and achieve better results. The network of the generator and discriminator of BSEGAN-SI selects a convolutional neural network as the core network layer of the whole structure, which is a fully convolutional network. The structure of the whole generator network and deep convolution generative adversarial networks (DCGAN) [29] is consistent with discarding the pooling layer. The pooling layer operation will cause part of the information loss of the speech features. For BC enhancement, it will reduce the enhanced BC speech quality. Normalization operation is a common method proposed to simplify operation. Generally, data are mapped to a relatively small range, which can facilitate data processing. Before network training, we preprocessed the extracted speech feature data, including normalization, to reduce the large difference between data, so that the model could be better trained. The convolution layer of the generator is realized by two-dimensional (2D) convolution. Gated CNN [30] adds gated linear units (GLUs) to the model to capture long-term memory, and the number of nonlinear calculations of this structure is reduced, which can accelerate the convergence speed of the model. We all adopt this structure in the encoder–decoder module and add a GLU layer to improve the model's ability to capture speech feature information. The output of the convolution layer becomes the following Formula (9),

$$O_{l+1} = (O_l \cdot W_l + c_l) \otimes \sigma(O_l \cdot V_l + d_l) \tag{9}$$

where $W$ and $V$ are different convolution kernels. That can be expressed as a "convolution layer output without nonlinear function" element-wise multiplied by "convolution layer output through sigmoid nonlinear activation function".

The middle of the encoder–decoder module is an intermediate layer composed of four convolutional layers, which are used to enable the model to capture more speech feature information. In our BC speech enhancement task, the entire GAN model adopts instance normalization (IN) [31] layer, which can effectively alleviate the gradient explosion problem. The encoder and decoder modules are similar in structure, which are constructed by 2D convolution, GLU, and IN layer stacking. The difference is that we use pixels in the decoder module pixel_2Dshuffler [32] to improve the effectiveness of the generated speech feature map and make it closer to the target AC speech feature map. In the image field, it has been verified that adding Pixel_Shuffler and channel dimension information in the up-sample module to fill pixels can generate more realistic images with ultra-high resolution.

The discriminator structure of BSEGAN-SI is shown in Figure 4. The discriminator model is a binary classification network. According to the typical classification network, the discriminator network we use has a nine-layer 2D convolution layer. The input speech feature map is continuously compressed through the convolution layer to convert the information into a multi-channel information feature map. This is also in order to reduce the impact of initialization parameters on the training results and speed up the training.

Except for the fact that the sigmoid activation function is used for the last layer of the network, the LeakyReLU activation function is used for the remaining layers of networks. In addition, the discriminator is similar to the generator in that the IN layer is added to normalize the output of the convolution layer. After the first convolution layer, the GLU is added to better obtain the speech feature information.

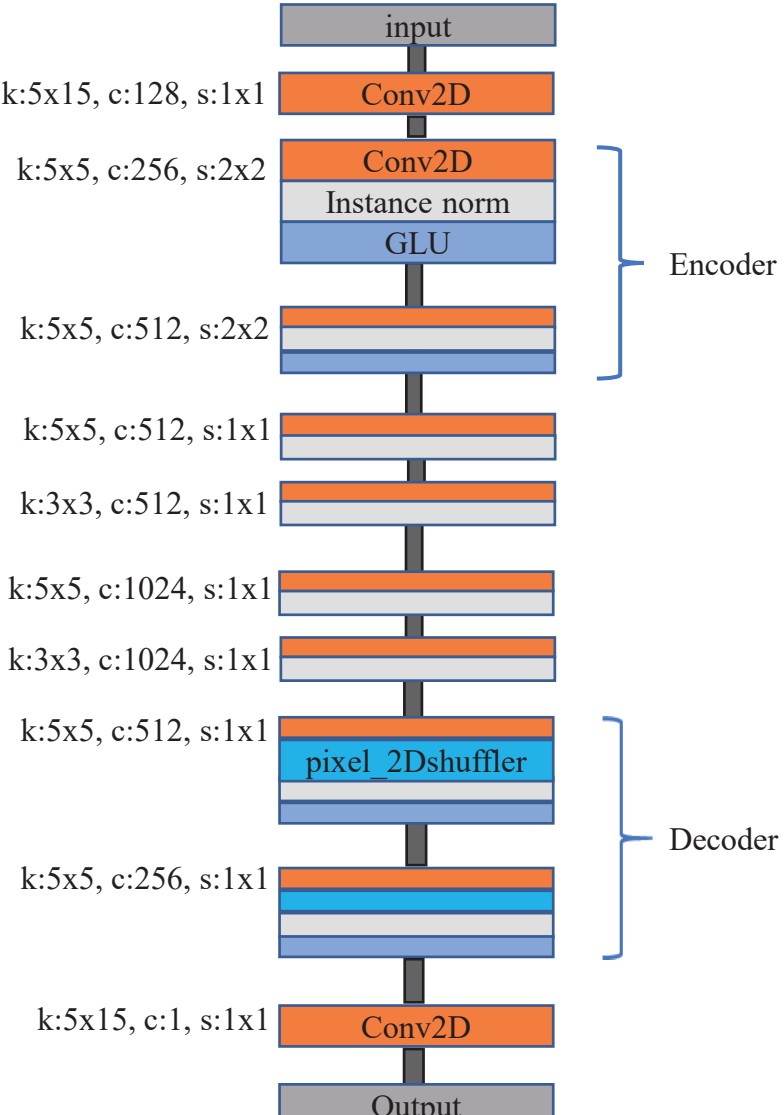

**Figure 3.** Generator structure of the BSEGAN-SI method.

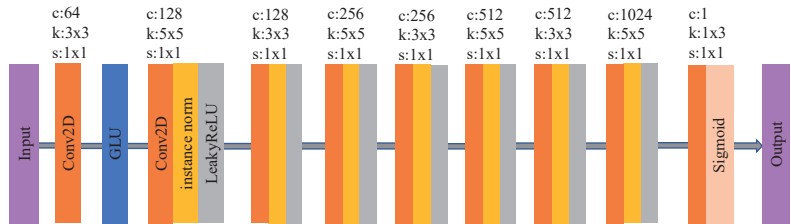

**Figure 4.** Discriminator structure of the BSEGAN-SI method.

### 3.3. Spectrum Distance Constraint

The proposed BSEGAN-SI method, under the confrontation training of the generator discriminator, enables the generator to continuously correct its output BC speech feature data to the data distribution similar to AC speech. In the experiment, we found that a single generator loss could not better establish a feature map closer to AC. To make the generated BC spectrum feature closer to the target AC spectrum data, we use the spectral distance constraint to narrow the gap. We choose to introduce the spectral distance constraint (L1 norm) in the BSEGAN-SI model due to the fact that it has been effectively applied in the field of image processing, and its formula is:

$$\mathcal{L}_{sd}(G) = \mathbb{E}_{s \sim S}[\|G(s) - t\|_1] \tag{10}$$

In summary, the target loss functions $\mathcal{L}$ of generator $G$ and discriminator $D$ of the proposed BSEGAN-SI are, respectively, as follows,

$$\mathcal{L}_{total}(G) = \mathcal{L}_{gen}^{adv} + \lambda_{sd} \mathcal{L}_{sd} \tag{11}$$

$$\mathcal{L}_{total}(D) = \mathcal{L}_{dis}^{adv} \tag{12}$$

where $\lambda_{sd}$ represents the trade-off coefficient of the loss function $\mathcal{L}_{sd}$. When the feature data generated by the generator gradually approaches the target data, the two-loss function Formulae (11) and (12) are used together to control the stability of the training. The network is iteratively trained to achieve convergence by optimizing the following Formula (13),

$$G^*, D^* = \arg \min_{G} \max_{D} \mathcal{L}_{total}(G, D) \tag{13}$$

Algorithm 1—the overall optimization process of the proposed BSEGAN-SI model in the training process is described in detail.

---

**Algorithm 1** BSEGAN-SI Model Training

---

**Input:** source BC speech features: $S$; target AC speech; features: $T$; Number of iterations: $N = 2 \times 10^5$.
**Output:** Trained BSEGAN-SI model.
    **for** n = 1 : N **do**
        Sample $s \in \mathbb{R}^{Q \times L_{BC}}$, $t \in \mathbb{R}^{Q \times L_{AC}}$ from $S$, $T$.
        $G(s) \leftarrow$ Generate enhanced features from $s$ with $G$.
        $D(G(s)) \leftarrow$ Classify real/fake features with $D$.
        Compute adversarial losses $\mathcal{L}_{gen}^{adv}$ and $\mathcal{L}_{dis}^{adv}$.
        Compute spectrum-distance loss $\mathcal{L}_{sd}$.
        Optimizing generator:
          $G \leftarrow \nabla \left( \mathcal{L}_{gen}^{adv} + \lambda_{sd} \mathcal{L}_{sd} \right)$
        Optimizing discriminator:
          $D \leftarrow \nabla \left( \mathcal{L}_{dis}^{adv} \right)$
    **end for**

---

## 4. Experiment

This section mainly verifies the effectiveness of the methods proposed in Section 3, we evaluate the speech quality of the enhanced speech using the above methods by objective and subjective evaluation indicators.

### 4.1. Experimental Dataset

We used three BC speech datasets to verify the model effect, namely, the TMHINT dataset proposed in [8], the AEUCHSAC&BC-2017 dataset proposed in [33], and the TMBCS dataset we established.

The TMHINT dataset is a balanced corpus consisting of 1920 statements. The statements in the dataset were recorded by six speaker participants in a standard recording room at a sampling rate of 44.1 kHz using AC and skull conduction microphones at the same time [8]. Each speaker was required to record 320 sentences, and the corpus of each sentence is composed of 10 Chinese characters.

For the AEUCHSAC&BC-2017 dataset, the adopted corpus consists of 1000 short sentences, and the sentence time after each recording was required to be 3–5 s. A total of 16 speakers were asked to read 200 sentences in the corpus randomly. In the professional recording and broadcasting room, the recording personnel needs to read 200 corpora in the corpus at random and use an AC microphone and throat microphone (TM) equipment to record voices. In this paper, we will use some open-source BC speech datasets obtained from this dataset for relevant experiments.

For the TMBCS dataset, 1600 Chinese corpora were selected as the corpus. The dataset was recorded by 10 speakers with a total of 3000 statements. Each speaker needs to use the AC microphone and throat microphone (TM) equipment to record 300 BC and AC speech, respectively, and the duration of each sentence is about 2–4 s. The dataset can be divided into two categories according to its purpose. The first category is suitable for BC speech enhancement of specific people; the second type is BC speech enhancement for speaker-independent systems. For the needs of this study, subset II is adopted.

### 4.2. Experimental Setup

The parameter configuration of the generator and discriminator in the BSEGAN-SI method proposed in this paper is listed in Tables 1 and 2. The initial learning rate of the generator network is set to $2e^{-4}$, and the initial learning rate of the discriminator network is set to $1e^{-4}$. In the preprocessing stage of this experiment, all BC and AC speech are downsampled to 16 kHz and use WORLD vocoder to extract fundamental frequency $F_0$, $Ap$ and $Sp$ features from the voice frame moving 5 ms every 25 ms [24]. Then transform $Sp$ features into $MCEPs$ for model training. To improve the randomness of each batch, instead of using the speech spectral feature sequence directly in order, we randomly intercepted a segment with a fixed length of 128 frames from the $MCEPs$ feature sequence. Use the Adam optimizer to regularize parameter $\lambda_{sd}$, which is set to 10. For each dataset, the training data and testing data are divided by a ratio of 4:1.

**Table 1.** Parameter setting of BSEGAN-SI generator.

| | Module | Kernel Size | Stride Size | Channels |
|---|---|---|---|---|
| | Conv2D | $5 \times 15$ | 1 | 128 |
| | Encoder | $5 \times 5$ | 2 | 256 |
| | | $5 \times 5$ | 2 | 512 |
| | Conv2D | $5 \times 5$ | 1 | 512 |
| Generator | Conv2D | $3 \times 3$ | 1 | 512 |
| | Conv2D | $5 \times 5$ | 1 | 1024 |
| | Conv2D | $3 \times 3$ | 1 | 1024 |
| | Decoder | $5 \times 5$ | 1 | 512 |
| | | $5 \times 5$ | 1 | 256 |
| | Conv2D | $5 \times 15$ | 1 | 24 |

**Table 2.** Parameter setting of BSEGAN-SI Discriminator.

|  | Module | Kernel Size | Stride Size | Channels |
|---|---|---|---|---|
|  | Gated-Conv2D | $3 \times 3$ | 1 | 64 |
|  | Conv2D | $5 \times 5$ | 1 | 128 |
|  | Conv2D | $3 \times 3$ | 1 | 128 |
|  | Conv2D | $5 \times 5$ | 1 | 256 |
| Discriminator | Conv2D | $3 \times 3$ | 1 | 256 |
|  | Conv2D | $5 \times 5$ | 1 | 512 |
|  | Conv2D | $3 \times 3$ | 1 | 512 |
|  | Conv2D | $5 \times 5$ | 1 | 1024 |
|  | Conv2D | $1 \times 3$ | 1 | 1 |

### 4.3. Effectiveness of the Proposed BSEGAN-SI Method

To confirm the effectiveness of the proposed BSEGAN-SI method in speaker-independent spectral enhancement for BC speech tasks, the methods used in this paper use speaker-independent BC data sets for model training. We selected the GMM [34], BLSTM [35], StarGAN [36], and CycleGAN model [16] as the comparison benchmark methods in this task.

Figure 5 is the spectrum diagram of BC speech and its corresponding AC speech. From the BC speech spectrum enhanced by multiple methods in Figures 6–8, it can be seen that the BSEGAN-SI method proposed in this paper is superior to the other benchmark methods. The traditional statistical method GMM has the worst enhancement effect on BC speech, with the least voiceprint loss and high-frequency component recovery in the speech spectrogram, and the spectrogram is fuzzy; compared with GMM, the spectrogram of BC speech enhanced by the BLSTM and CycleGAN models is better, but some voiceprint details are still missing in the middle and high-frequency component of the speech. After the StarGAN model is enhanced, the voice print of BC speech is lost, and the last noise area appears in the spectrogram. Our method can better recover the missing components of BC. The spectral details are clearer and most similar to the target AC speech spectrum. The enhanced speech quality has been better improved, which shows the effectiveness of BSEGAN-SI in speech enhancement tasks.

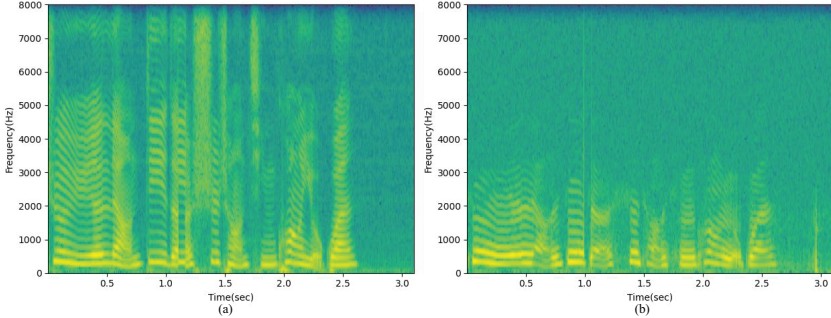

**Figure 5.** (**a**) AC speech spectrogram. (**b**) BC speech spectrogram.

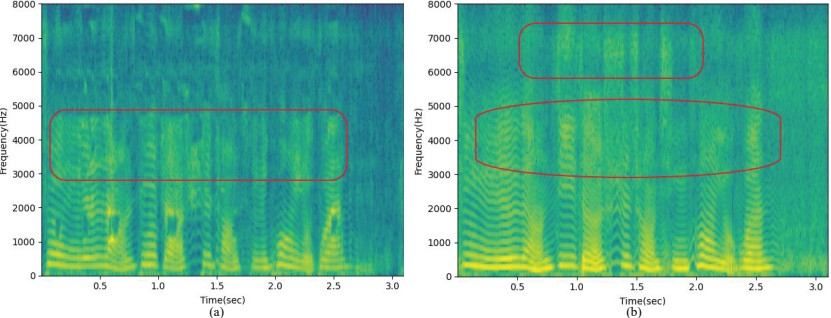

**Figure 6.** (**a**) Enhanced BC speech spectrogram by GMM. (**b**) Enhanced BC speech spectrogram by BLSTM.

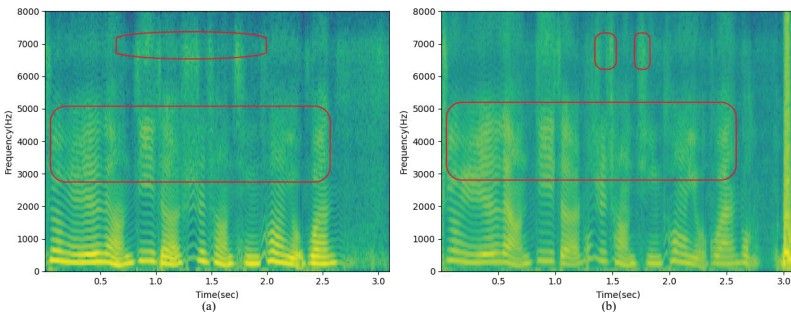

**Figure 7.** (**a**) Enhanced BC speech spectrogram by CycleGAN. (**b**) Enhanced BC speech spectrogram by StarGAN.

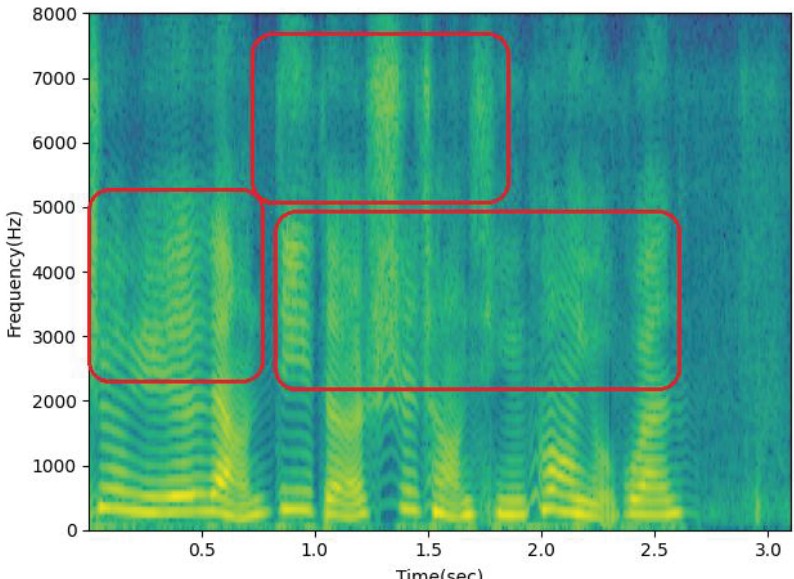

**Figure 8.** Enhanced BC speech spectrogram by BSEGAN-SI.

*4.4. Objective Evaluation*

The short-time objective intelligibility (STOI) [37], P.563 [38] and log-spectral distortion (LSD) [39] was used to objectively evaluate the quality of the enhanced BC speech to evaluate the effectiveness of the proposed BSEGAN-SI method. In STOI and LSD, the distortion value of some aspects of speech before and after enhancement is calculated to judge whether it is good or bad. Short-time objective intelligence (STOI) is a digital measurement method based on the frequency domain, which aims to evaluate the intelligibility of enhanced BC speech. The evaluation score range is 0–1. The larger the score, the higher the enhanced speech intelligibility.

In the objective evaluation index of speech enhancement system, logarithmic spectral distance is widely used because of its high degree of objectivity. The core idea of the LSD algorithm is to evaluate the approximation of two speech signals by calculating the logarithmic spectrum distance of clean AC speech and enhanced BC speech. In this paper, LSD between BC speech and normal speech is used as the objective evaluation index of the BC speech enhancement system. LSD is defined as follows:

$$\mathrm{LSD} = \frac{1}{N} \sum_{i=1}^{N} \sqrt{\frac{1}{M/2+1} \sum_{j=0}^{M/2} \left( 10 \log_{10} \left| S_{i,j} \right| - 10 \log_{10} \left| X_{i,j} \right| \right)^2} \tag{14}$$

where $N$ is the number of speech frames, $M$ is the number of DFT sample points, and $X_{i,j}$ and $S_{i,j}$ represent the STFT cepstrum of pure speech and enhanced speech in the $i$th and $j$th frames, respectively. STFT cepstrum is a description of the relationship between spectrum

and time. By comparing the spectrum of two kinds of speech, we can clearly see the effect of the speech enhancement algorithm. The smaller the LSD value is, the more similar the log spectrum between the enhanced BC speech and its corresponding pure AC speech is, and the closer the enhanced BC speech quality is to the pure AC speech quality.

P.563 is a single-ended speech quality objective evaluation standard proposed by ITU-T [38]. The measurement process of p.563 consists of three stages: the preprocessing stage, the distortion estimation stage, and the perceptual mapping stage. It does not need to use pure speech as a condition when evaluating BC speech quality. It can directly output the smoothness score of the signal [40], which has higher usability. The grades at which the bothersome or perceptual focus was found by analyzing the auditory experiments are presented in Table 3. It can be seen from Figure 9 that we have conducted a P.563 evaluation to enhance the effectiveness of the proposed BSEGAN-SI method. Considering the difference between BC and clean AC speech, the single-ended indicator P.563 is particularly suitable for objective evaluation of BC speech quality obtained in BC speech enhancement tasks.

**Table 3.** P.563 distortion evaluation.

| Grade | Distortion Classification |
|-------|---------------------------|
| 1 | High-level background noise |
| 2 | The signal is interrupted |
| 3 | Signal-related noise |
| 4 | Machine speech |
| 5 | Common unnatural voice |

As shown in Table 4, the STOI values of the enhanced BC speech obtained by the comparison model GMM, BLSTM, StarGAN, CycleGAN, and the BSEGAN-SI method proposed in this paper are listed. It can be seen that the STOI score of the enhanced BC speech obtained by our method is the highest, indicating that the enhanced BC has better quality and higher clarity. Table 5 lists the LSDs of the enhanced BC speech obtained by different methods. The spectral distortion of the enhanced BC speech obtained by our method is the lowest, which indicates that the logarithmic spectral similarity between the enhanced BC speech and the clean AC speech is higher, and the information component of the enhanced BC speech spectrum is closer to the target AC speech spectrum. Table 6 lists the P.563. The method we proposed has the highest score, indicating that the voice quality is the best. In order to better observe the score of the objective evaluation index of the enhanced voice obtained by enhancing the BC speech of the three data sets through various methods, we convert all the objective evaluation results into a line chart. As shown in Figures 9–11, for three different speaker-independent BC speech datasets, our method achieves the best result of enhanced BC speech. Therefore, this analysis further strengthens the effectiveness of our proposed BSEGAN-SI method.

**Table 4.** STOI evaluation of the enhanced BC speech obtained by different methods (double-ended indicators).

| Datasets | BC | GMM | BLSTM | CycleGAN | StarGAN | SIGAN-BCSE |
|----------|------|------|-------|----------|---------|------------|
| AEUCHSAC&BC-2017 | 0.620 | 0.656 | 0.771 | 0.778 | 0.730 | 0.809 |
| TMHINT | 0.686 | 0.717 | 0.813 | 0.821 | 0.749 | 0.842 |
| TMBCS | 0.661 | 0.718 | 0.780 | 0.773 | 0.743 | 0.810 |

**Table 5.** LSD evaluation of the enhanced BC speech obtained by different methods (double-ended indicators).

| Datasets | BC | GMM | BLSTM | CycleGAN | StarGAN | SIGAN-BCSE |
|----------|-------|-------|-------|----------|---------|------------|
| AEUCHSAC&BC-2017 | 1.779 | 1.462 | 1.163 | 1.130 | 1.303 | 1.119 |
| TMHINT | 1.235 | 1.075 | 0.998 | 0.995 | 1.199 | 0.986 |
| TMBCS | 1.794 | 1.513 | 1.119 | 1.115 | 1.232 | 1.101 |

**Table 6.** P.563 evaluation of the enhanced BC speech obtained by different methods (single-ended indicator).

| Datasets | BC | GMM | BLSTM | CycleGAN | StarGAN | SIGAN-BCSE |
|----------|------|------|-------|----------|---------|------------|
| AEUCHSAC&BC-2017 | 2.912 | 2.988 | 4.320 | 4.418 | 4.205 | 4.433 |
| TMHINT | 1.948 | 2.322 | 4.276 | 4.375 | 3.937 | 4.515 |
| TMBCS | 2.689 | 2.893 | 4.000 | 3.925 | 3.830 | 4.207 |

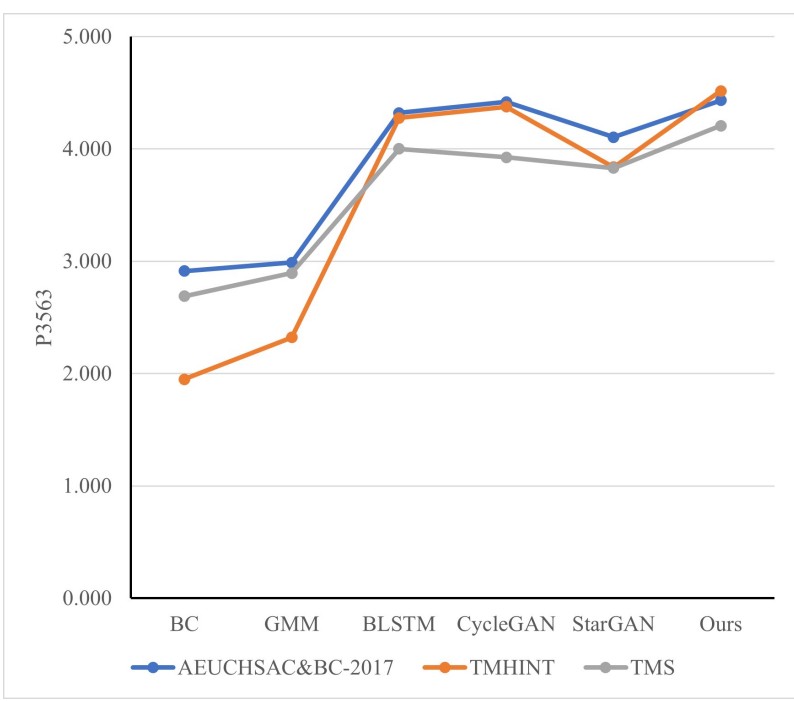

**Figure 9.** P.563 scores of enhanced BC speech obtained by different methods.

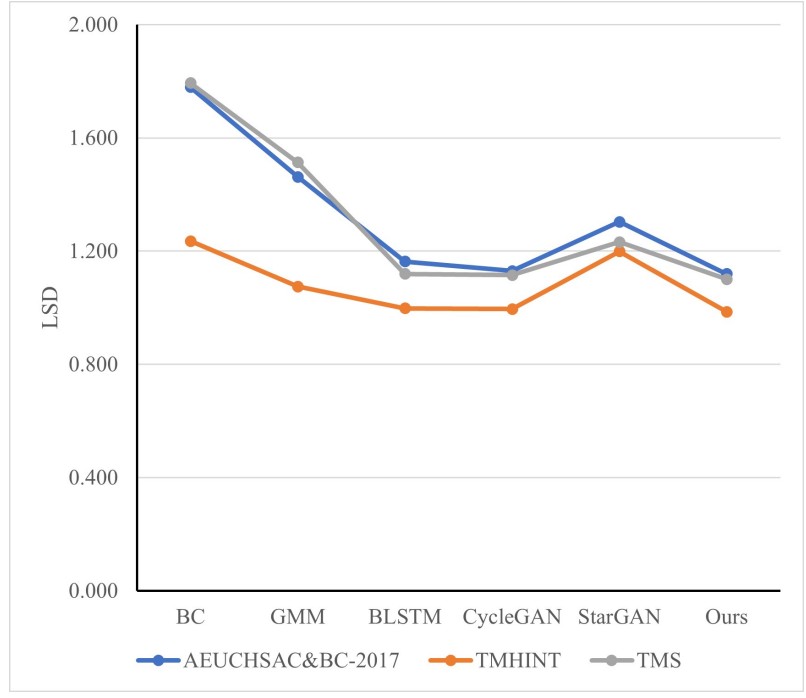

**Figure 10.** LSD scores of enhanced BC speech obtained by different methods.

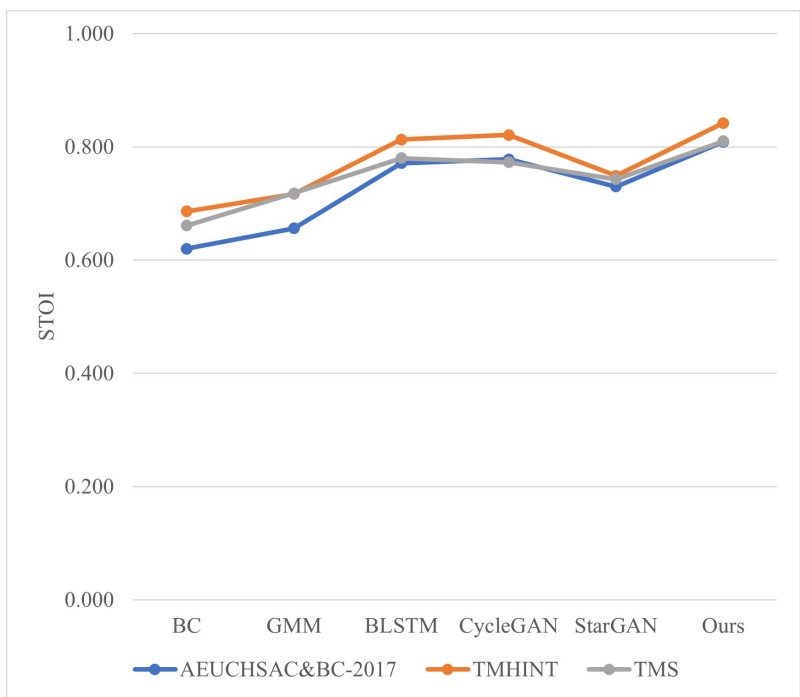

**Figure 11.** STOI scores of enhanced BC speech obtained by different methods.

### 4.5. Subjective Evaluation

The subjective evaluation method is different from the above method, which is based on the evaluation of people. In this paper, the mean opinion score (MOS) [41] was used to evaluate the auditory perception of the enhanced BC speech. A total of 20 subjects (10 male and 10 female, with good subjective auditory perception function) participated in the subjective test. During the evaluation, the BC speech before and after the enhancement is played at random, and then the subjects subjectively score the spoken words. The score range of MOS in this evaluation is 0–5 points. The higher the score, the better the subjective auditory quality of enhanced BC speech. The MOS evaluation criteria are shown in Table 7 below.

**Table 7.** Sound quality scoring standard.

| Score | Quality Level | Subjective Perception |
|-------|---------------|-----------------------|
| 0–1 | Bad | Very uncomfortable and intolerable |
| 1–2 | Poor | Uncomfortable but tolerable |
| 2–3 | Fair | Perceived distortion and uncomfortable |
| 3–4 | Good | Slight distortion but not annoying |
| 4–5 | Excellent | Imperceptible |

The calculation method of the MOS score is shown as follows,

$$Score_{MOS} = \frac{1}{M \times N} \sum_{i=1}^{M} \sum_{j=1}^{N} score_{i,j} \tag{15}$$

where M represents the number of people participating in the evaluation, and N represents the number of speech pieces evaluated by each person. $score_{i,j}$ is the *ith* evaluation.

In this experiment, we tested the $Score_{MOS}$ of five BC enhancement methods, respectively. The $Score_{MOS}$ of all models is intuitively shown in Figure 12. The score of our method is better than other methods. It can be seen from the results in the figure that the enhanced BC speech enhanced by our proposed BC enhancement method has higher clarity and naturalness.

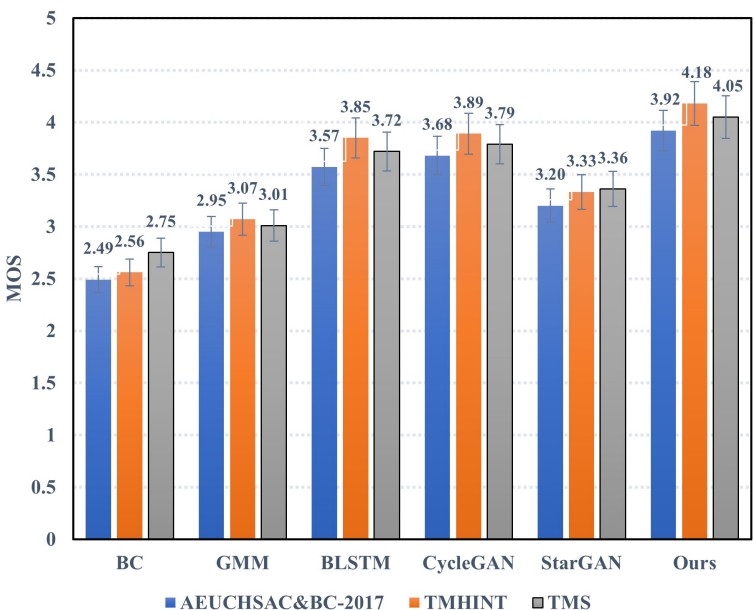

**Figure 12.** MOS with different BC speech enhancement methods.

## 5. Conclusions

In this paper, we introduce a BSEGAN-SI method for speaker-independent spectral enhancement for BC speech. By establishing feature mapping between speech, we can better enhance the auditory effect of speech. This method uses spectral distance to constrain the difference between the BC speech spread spectrum and the AC speech spectrum so that the generated enhanced BC speech spectrum is closer to the AC speech spectrum. In addition, the parameters and complexity of the model are low, which can achieve a better enhancement effect through short-time training. The above results show that this method can better recover the lost components of the BC speech spectrum, obtain a more complete spectrum structure similar to the AC speech, improve the auditory perception and articulation quality of BC speech, and help improve the quality of speech communication in poor environments. Furthermore, after the training of the model, the enhancement of BC speech does not need the help of AC speech information. As a future extension, the model can be used as the final system for real-time operation. If you want to obtain a program code for research, please email e20201132@stu.ahu.edu.cn.

**Author Contributions:** Conceptualization, L.C., software, L.C. and H.W.; methodology, L.C. and Y.D.; validation, Y.D. and J.Z.; resources, L.T. and J.Z.; writing—original draft preparation, L.C.; writing—review and editing, L.C., J.Z., Y.D., and H.W.; supervision, L.T. and H.W. All authors have read and agreed to the published version of the manuscript.

**Funding:** This research was funded by the National Natural Science Foundation of China Joint Fund Key Project (U2003207), National Natural Science Foundation of China (61902064), Natural Science Foundation of Anhui Province (No. 1908085MF209), Key Projects of Natural Science Foundation of Anhui Province Universities (No. KJ2018A0018).

**Data Availability Statement:** Not applicable.

**Conflicts of Interest:** The authors declare no conflict of interest.

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
