# Peer review of "Speaker-Independent Spectral Enhancement for Bone-Conducted Speech"

_algorithms, doi:10.3390/a16030153_

Round 1
Reviewer 1 Report
The paper aims at improving the enhancement effect of BC speech for speaker-independent to recover the missing components of BC speech. In addition, the method adds the training of the spectral distance constraint model, and finally uses the enhanced model completed by the training to reconstruct BC speech. The study claimed that this method is superior to the existing methods.
The paper is well organized and written that provides the readers with BC speech recognition, the current limitations and the solution proposed to solve the current limitations of BC speech recognition. However, there are some improvements that the authors need to do, to make the paper suitable for publication.
1. The authors failed to provide the full term for many abbreviations when they were first introduced. Examples are
a. BLSTM at line 81
b. BSEGAN-SI at line 123
c. DCGAN at line 269
d. TMHINT at line 331
e. TMBCS at line 332
2. More explanations need to be provided on the speech databases used for the experiments (the TMHINT, AEUCHSAC&BC-2017, and TMBCS). Information such as dataset size, speakers, and the data for training and testing for each database needs to be provided.
3. P.563, the single-ended speech quality objective evaluation needs to be explained further.
4. Figure which is not suitable for publications:
a. Figure 5, the spectrogram images are too small for readers.
Reviewer 2 Report
This paper considers processing of sound signals obtained from bone conduction as they are to be improved by air conduction. As such it would appear useful and to be publishable. However some things would be helpful to a reader: 1. The English overall is good but there are some problems, such as with line 34 of page 1 with "precise" which should probably be "precisely". So English could be. checked overall 2. The Abstract is long for what is needed which is just what is relevant to what is new in the paper. 2. There are lots of abbreviations most of which are well spelled out but the key one of BSEGAN-SI is not found? 3. As seemly understood, the bottom of Fig. 1 is the end result once trained with the top of the figure. If so, this should be made clearer. Also it would help to show where some of the later figures fit into it and the names in the two figures be the same (such as Generator and Discriminator which do not appear in Fig. 1). 4. Actually a figure showing how this system fits on the human showing a bone (with sensor) - the processing of this paper - and the microphone (supposedly in an ear). 5. It is not clear if the data used is for bone behind the ear or for the jaw (at least without digging into the data sources). 6. The training should use a lot of resources to handle sufficient AC information. As a future extension could not real time AC be used so that the full Figure 1 would be the end system operating all in real time, rather than just the bottom part?
Round 2
Reviewer 2 Report
The improvements are very helpful. It would be even more helpful if the comments to the reviewer on real time use and ear vesus jaw were discussed in the conclusions.
Also the programs used should be made available to a reader with a statement, best in the conclusions, as to how they can be obtained. Especially this is important since this is a system for eventual human use.
